# Effects of Different Land Use Types and Soil Depth on Soil Nutrients and Soil Bacterial Communities in a Karst Area, Southwest China

**Yuke Li** [1,†], **Jiyi Gong** [1,†], **Jie Liu** [1], **Wenpeng Hou** [2,3], **Itumeleng Moroenyane** [4], **Yinglong Liu** [2,3], **Jie Jin** [3], **Jie Liu** [2,3], **Han Xiong** [1], **Chen Cheng** [2,3], **Kamran Malik** [2,3], **Jianfeng Wang** [2,3,5,6,*] and **Yin Yi** [1,*]

1   Key Laboratory of National Forestry and Grassland Administration on Biodiversity Conservation in Karst Mountainous Areas of Southwestern China, Guiyang Normal University, Guizhou 550025, China; 201407072@gznu.edu.cn (Y.L.); 201307048@gznu.edu.cn (J.G.); liujie791204@126.com (J.L.); wnyxjj@126.com (H.X.)

2   State Key Laboratory of Grassland Agro-Ecosystems, Center for Grassland Microbiome, Lanzhou University, Lanzhou 730000, China; houwp19@lzu.edu.cn (W.H.); liuyl2020@lzu.edu.cn (Y.L.); jieliu@lzu.edu.cn (J.L.); chengch20@lzu.edu.cn (C.C.); malik@lzu.edu.cn (K.M.)

3   College of Pastoral Agriculture Science and Technology, Lanzhou University, Lanzhou 730000, China; jinj2015@lzu.edu.cn

4   Institut National Recherche Scientifique, Centre Armand-Frappier Santé Biotechnologie 531 Boulevard des Prairies, Quebec City, QC H7V 1B7, Canada; Itumeleng.Moroenyane@inrs.ca

5   Collaborative Innovation Center for Western Ecological Safety, Lanzhou University, Lanzhou 730000, China

6   State Key Laboratory of Plateau Ecology and Agriculture, Qinghai University, Xining 810016, China

*   Correspondence: wangjf12@lzu.edu.cn (J.W.); yiyin@gznu.edu.cn (Y.Y.)

†   These authors contributed equally to this work.

**Abstract:** To reveal the effect of the interactions between soil depth and different land use types on soil nutrients and soil bacterial communities in a karst area, fifty soil samples from five different karst land use types in Huajiang town, Guizhou province, Southwest China were collected, and the soil bacteria were analyzed using high-throughput absolute quantification sequencing. Our results showed that land use types (LUT) and soil depth (SD) significantly influenced the content of soil organic carbon (SOC), total nitrogen (TN), total phosphorus (TP), nitrate nitrogen (NN), ammonium nitrogen (AN) and available soil phosphorus (AP), and pH; further, the interaction of LUT and SD also significantly influenced SOC, NN, NA, AP, and pH. In addition, LUT clearly impacted the Chao1 and Shannon indexes, but, SD and LUT * SD markedly affect Chao1 and Shannon index, respectively. All the soil bacterial communities were significantly different in the five different five land use types according to PERMANOVA. Importantly, Acidobacteria and Proteobacteria were the predominant phyla at soil depths of 0–20 cm and 20–40 cm among all the LUTs. At 0–20 cm, TN, AN, and SOC exerted a strong positive influence on Acidobacteria, but NN exerted a strong negative influence on Acidobacteria; at 20–40 cm soil, TN and AN exerted a strong positive influence on Acidobacteria; TP exerted no marked influence on any of the phyla at these two soil depths. At 0–20 cm of soil depth, we also found that Chao1 index changes were closely related to the TN, SOC, AN, and NN; similarly, Shannon index changes were significantly correlated to the AN, TN, and SOC; the PCoA was clearly related to the TN, SOC, and AN. Interestingly, at soil depth of 20–40 cm, Chao 1 was markedly related to the TN and pH; Shannon was markedly correlated with the SOC, TP, AN, and AP; and the PCoA was significantly correlated with the TN and pH. Our findings imply that soil nutrients and soil bacteria communities are strongly influenced by land use types and soil depth in karst areas.

**Keywords:** karst areas; soil depth; land use types; soil nutrients; soil bacterial community and diversity; high-throughput absolute abundance quantification

## 1. Introduction

Karst topography is widely distributed globally and accounts for about 15% of the global land surface [1]. Although the karst topography of Southwest China covers a range of about 540,000 km$^2$, making it one of the three largest karst regions on a global scale, the available land area in karst areas is limited [2–4]. Karst ecosystems are some of the most vulnerable areas and respond very quickly to human impact [5], and karst ecosystems are difficult to restore when they are disturbed [6]. The land available for cultivation is limited by the sufficient depth of soil distribution between rock outcrops in karst regions. Therefore, the rational use of land and soil depths in karst areas is critical. Recent reports highlight that land use type, revegetation process, and rocky karst desertification influence the structure and diversity of microbial communities [7–10]. However, the coupled influence of land use and soil depth on the composition and structure of soil microbial communities remains unknown.

Soil microorganisms play a central role in the restoration of ecosystems by improving soil nutrients, soil functions and vegetation restoration [11–16]. The decomposition of soil organic matter [17] and the release of biologically active nitrogen and available phosphorus for plant growth are governed by microbes [14]. In the past decade, Guizhou provincial government has encouraged farmers to plant different cash crops, which can, on the one hand, prevent rocky desertification, and increase farmers' incomes on the other, leading to change in land use types and below-ground dynamics. These fluctuations, in turn, exert profound direct and indirect effects on soil microbes. Studies have demonstrated that changes in land use types greatly influence the structure and diversity of soil microbial communities [18,19].

Agricultural management, including irrigation, fertilization, and land use types, is a key factor that can degrade soil ecosystem and function and also influence the biodiversity of terrestrial ecosystems [20–22]. Land use practices influence soil health and nutrient status [23]. In karst regions, soil microbe abundance and diversity in rice field soil are markedly lower than maize and citrus field soil [9]. Differences in the abundance and diversity of soil microbial communities in karst peak-cluster depressions correspond to different land use types (primary forest, secondary forest, farmland, forest plantations, scrubland and grassland) [24]. Soil microbes responded differently to the change in land use type, which led to significant nutrient cycling changes [25]. However, soil chemical property effects were more significant than the effects of land use type on microbial community structures in some ecosystems [26]. Microbial community structures are susceptible to different land use types; however, whether these influences extend down the soil profile is still unknown.

The different relative abundance of soil bacterial communities can be a potential biological indicator of environmental status [27]. Microbial abundance and diversity typically decrease with soil depth [28,29]. Despite studies highlighting changes in microbial community structure along soil depths, few have shown detailed taxonomic and phylogenetic features of these trends [30,31]. Despite these results, our understanding of soil microbial function for ecosystem stability remains limited, mainly concerning community structure changes and diversity of soil depth among different land use systems. In general, microbial diversity decreases with soil depth caused multiple alterations in soil properties [32–34]. Soil nutrient cycling is a microbial-mediated biological process that can be predicted on a large spatial scale according to abiotic factors, such as soil nutrient availability and soil pH [35].

The diversity and composition of soil bacterial communities was significantly affected with cultivation disturbance in karst areas [36]. Soil microbes are often limited by carbon, but Schimel and Weintraub [37] reported that phosphorus and nitrogen were also limiting for microbial growth and community structure [37–39]. Microbes at deeper soil depths play central roles in soil development, nutrients, and carbon storage potential and feature lower turnover rates than top-surface communities [40,41]. Comparing uncharacterized

soil depth-dependent responses to land use type changes represents an understudied aspect of soil microbial function and ecology [34].

Soil microbes in the subsoil exert an essential influence on ecosystem biochemistry, soil formation and maintaining groundwater quality [32,42]. The abundance of Actinobacteria and Proteobacteria in the topsoil increased when using organic fertilizer and chemicals, but Acidobacteria abundance decreased [43]. Microbial mineralization was more susceptible to subsoil amendments than in topsoil [29]. The different microorganism genus formed complex and powerful interaction networks in ecosystems. Clearing the interactions between microbes is essential to investigate the intricacy of soil's chemical process and biological function [44]. Nevertheless, microbial taxa's response at different soil depths and land use in the karst areas of southwest China remain unknown.

The structure and diversity of soil bacterial communities were strongly affected by soil physical and chemical properties, such as soil texture [45,46], soil pH [47,48], and nutrient availability [49]. Soil heterogeneity caused by fertilization significantly influenced the abundance and diversity of bacterial communities at different soil depths [43,50]. It was also shown that different land use types feature different soil nutrient requirement for plants growth [51,52].

However, the impact of different land use types on the diversity and structure of soil bacterial communities and soil nutrients at different soil depths has not yet been investigated in the karst areas of Southwest China. Therefore, our aim was to assess (a) whether soil nutrients were influenced by land use types and soil depths, (b) whether the structure and diversity of soil bacterial communities were influenced by land use types and soil depths, (c) and investigate the relationship between the alpha diversity and structure of soil bacterial communities and soil the nutrients of different soil depths in different land use types.

## 2. Materials and Methods

### 2.1. Study Site Description and Soil Sampling

The present study was carried out in a typical karst region of Huajiang tow, which is located in the southwest of Guanling Buyi and Miao Autonomous County, Guizhou Province. The annual mean temperature and the annual mean rainfall are about 17 °C and 1200 mm, respectively, and the frost-free period is about 288 days. The different land use types include *Zanthoxylum planispinum* land, *Hylocereus* spp. land, *Zea mays* land, grassland (the main species are *Themeda japonica*) and secondary forest land (the main species are *Liquidambar formosana*). Detailed information on the different land use types included in supplementary Table S1. In November 2019, soil samples from five different land use types were collected at two depths, 0–20 cm and 20–40 cm, with the topsoil removed (1–2 cm) using a clean stainless steel shovel placed in a plastic bag on ice until transfer to the laboratory, and with visible roots and stones removed. Five plots that were 5 m apart were sampled at each site at the 0–20 cm and 20–40 cm soil depths. Each plot was pooled from four subplots (1 m²) taken inside the plot at 0–20 cm and 20–40 cm soil depths. In order to prevent the influence of plant roots and other visible debris, all soil samples were sifted through a 2 mm mesh before laboratory analysis. A portion of each soil sample was placed in a 50 mL sterile centrifuge tube and flash-frozen in liquid nitrogen. The sterile centrifuge tubes were stored at −80 °C until soil DNA extraction. The remainder of the soil from each sample was air-dried to determine soil nutrients and pH.

### 2.2. Soil Nutrients Analyses

The content of soil organic carbon (SOC) and available soil phosphorus (AP) was analyzed with the method described by Nelson and Sommers [53]. Soil total phosphorus (TP), total nitrogen (TN), nitrate nitrogen (NN) and ammonium nitrogen (AN) were analyzed according to Zhao et al. [54]. Distilled water and soil samples were mixed in a ratio of 1:2.5, and the samples were shaken for 35 min. Subsequently, soil pH was determined with a pH meter.

*2.3. High-Throughput Absolute Abundance Quantification 16S-seq*

In total, 50 soil samples were collected from the 0–20 cm and 20–40 cm depth of 5 different land use types, and DNA from 350 mg soil was extracted by employing a Power Soil DNA Kit (MoBio, Carlsbad, CA, USA) according to the manufacturer's instructions. Furthermore, soil DNA were sent to Genesky Biotechnologies Inc., Shanghai, 201315 (China) for a high-throughput absolute quantification 16S rRNA gene amplicon sequencing by an Illumina MiSeq 2 × 250 bp sequencer. In the studies by Tkacz et al. (2018), Smets et al. (2016), Mou et al. (2020), and Jiang et al. (2019) [55–58], we can see the illustration of the high-throughput absolute quantification 16S-seq (HAQS). Briefly, the integrity of genomic DNA was detected through agarose gel electrophoresis, and the purity and concentration of genomic DNA were detected through the Nanodrop 2000 and Qubit 3.0 Spectrophotometer. Multiple spike-ins with identical conserved regions to natural 16S rRNA genes and variable regions replaced by random sequence with ~40% GC content were artificially synthesized. Next, an appropriate proportion of spike-ins mixture with known gradient copy numbers were added to the sample DNA. Multiple spike-ins with identical conserved regions to 16S rRNA gene and variable regions replaced by random sequence with ~40% GC content were artificially synthesized, and an appropriate mixture with known gradient copy numbers of spike-ins was then added to the sample DNA. Next, according to the description by Cai et al. [59], the amplified libraries were generated by amplifying with primers 515F (GTGCCAGCMGCCGCGG) and 907R (CCGTCAATTCMTTTRAGTTT) of the 16S rRNA gene.

*2.4. 16S rRNA Gene Sequence Analysis*

The original sequencing data were analyzed according to the method described by Huang et al. [60]. The primer and adaptor sequences were removed using the Mothur pipeline (Edition 1.39.3, https://www.mothur.org/ accessed on 30 October 2022) and TrimGalore (version 0.4.5, Babraham Bioinformatics, Cambridge, UK), respectively. After pair-end reads merging and filtering, low quality reads (<200 bp, ambiguous base calls > 0, average quality score < 20) were discarded. The rest sequences were clustered into OTUs with a similarity level at 97%, and mother (v.1.39.3) was used to assign taxonomy with RDP database. The spike-in sequences were then filtered out, and the readings were calculated. OTUs were then annotated, and the readings of each peaking OTU were counted. Finally, the standard curve of read amount compared to DNA copy peaks was established. The absolute copy number of bacterial OTUs may be calculated by the reading count of corresponding bacterial OTUs.

*2.5. Statistical Analyses*

The soil bacterial alpha diversity (Chao1 and Shannon indices) calculations were performed using R (version 3.2.2). The heatmap, PCoA, RDA, Spearman's rank correlation analysis, variation partition analysis (VPA), and Structural Equation Modeling (SEM) were carried out in R (version 3.2.2). In addition, permutational multivariate two-way analysis of variance (PERMANOVA) was carried out to make the estimation of the significant differences of bacterial community between land use type and soil depth based on Bray-Curtis distances. Two-way ANOVA was used to determine the effects of land use types (LUT) and soil depth (SD) on Chao1, Shannon, phyla levels of soil bacteria, SOC, TN, TP, NN, AN, AP and pH with SPSS (version 17.0). Significant differences between depths of 0–20 cm and 20–40 cm for Chao1, Shannon, SOC, TN, TP, NN, AN, AP, and pH in the five different land use types were carried out at $p < 0.05$ (independent *t*-tests) by Spss.

## 3. Results

*3.1. Soil Nutrient Status of the Different Depth under the Five Different Land Use Types*

We found that the interaction of land use types (LUT) × soil depth (SD) influenced SOC ($p < 0.001$, Table 1), NN ($p < 0.001$, Table 1), AN ($p < 0.001$, Table 1), AP ($p < 0.001$, Table 1), and pH ($p < 0.001$, Table 1). Further, we found that the SOC content at the

0–20 cm depth decreased by 68.7% compared to 20–40 cm depth under *Zanthoxylum planispinum* land (Figure 1B). The SOC content at 0–20 cm increased by 82.3% compared to the 20–40 cm depth under *Hylocereus* spp. (Figure 1B). Interestingly, the content of TN at the 0–20 cm depth was significantly enhanced, by 48.9%, compared to 20–40 cm depth under *Zanthoxylum planispinum* land (Figure 1C). The NN content at the 0–20 cm depth decreased by 97.6% compared to 20–40 cm depth under forest land. However, the NN content at the 0–20 cm depth increased by 1640.0%, 409.4%, and 300.0% compared to 20–40 cm depth under *Zanthoxylum planispinum* land, Hylocereus spp. Land, and *Zea mays* land, respectively (Figure 1E). The AN content at the 0–20 cm depth decreased by 56.6%, 31.8%, and 39.8% compared to 20–40 cm depth under grassland land, *Zanthoxylum planispinum* land, and *Hylocereus* spp. land (Figure 1F). The AP content at the 0–20 cm depth decreased by 68.4% and 70.3% compared to the 20–40 cm depth under grassland and *Hylocereus* spp. Land, respectively (Figure 1G). The soil pH at the 0–20 cm depth increased by 5.7% and 45.5% compared to 20–40 cm depth under forest land and *Hylocereus* spp., respectively, but the pH at the depth of 0–20 cm decreased by 4.0% compared to the 20–40 cm depth under *Zea mays* land (Figure 1H). Lastly, the TP content did not vary significantly across different LUTs and soil depths (Figure 1D).

**Table 1.** Results of two-way ANOVA for the effects of land use type (LUT) and soil depth (SD) on SOC, TN, TP, NN, AN, AP, and pH, F: F-value.

| Treatment | df | SOC | | TN | | TP | | NN | | AN | | AP | | pH | |
|---|---|---|---|---|---|---|---|---|---|---|---|---|---|---|---|
| | | F | $p$ | F | $p$ | F | $p$ | F | $p$ | F | $p$ | F | $p$ | F | $p$ |
| LUT | 4 | 58.7 | <0.0001 | 18.5 | <0.0001 | 3.2 | 0.023 | 13.2 | <0.0001 | 174.5 | <0.0001 | 22.6 | <0.0001 | 90.0 | <0.0001 |
| SD | 1 | 46.3 | <0.0001 | 14.9 | 0.0004 | 5.2 | 0.028 | 14.7 | 0.0004 | 127.4 | <0.0001 | 40.2 | <0.0001 | 76.6 | <0.0001 |
| LUT × SD | 4 | 59.1 | <0.0001 | 1.2 | 0.309 | 1.0 | 0.399 | 18.3 | <0.0001 | 64.4 | <0.0001 | 14.5 | <0.0001 | 75.3 | <0.0001 |

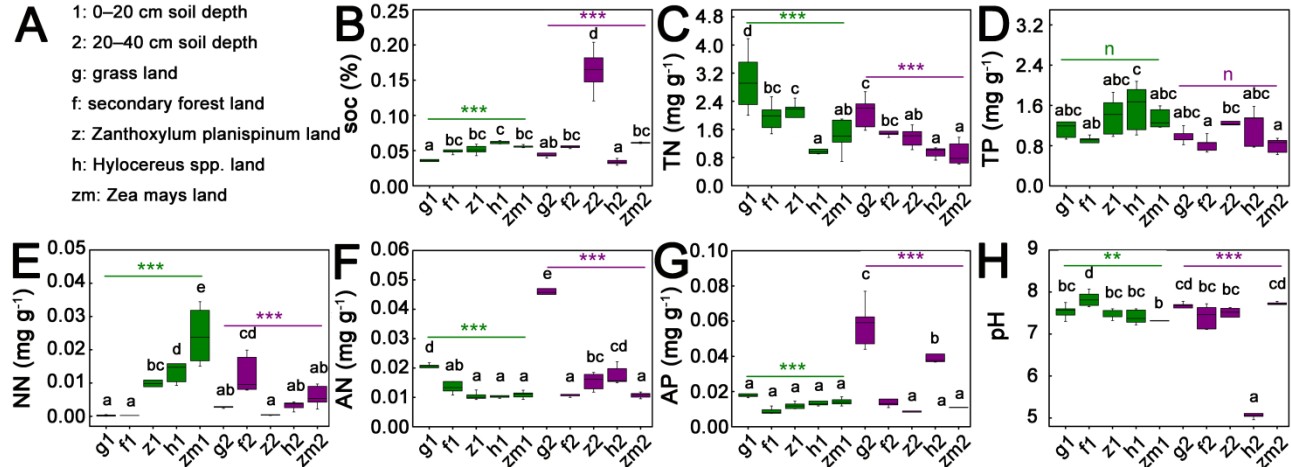

**Figure 1.** Effect of the different land use types on soil nutrients in 0–20 cm and 20–40 cm soil depth. (**A**) Description of abbreviations, (**B**) SOC content, (**C**) TN content, (**D**) TP content, (**E**) NN content, (**F**) AN content, (**G**) AP content, (**H**) pH. The different letters are significantly different with ANOVA followed by Duncan's multiple range test at the level of $p < 0.05$. ** and *** above green line and purple line indicate obvious difference at $p < 0.01$ and $p < 0.001$ at 0–20 cm and 20–40 cm soil depths, respectively.

### 3.2. Alpha Diversity Patterns

A total of 19,732,834 reads remained, which represented 19,352 bacterial OTUs after removing singleton OTUs, ambiguous, short and low-quality reads. The rarefaction curve is shown in supplementary Figure S1. Our results found that land use types (LUT) and soil depth (SD) exerted a clear effect on Chao 1 ($p < 0.001$; $p = 0.014$) (Figure 2A), but the interaction of LUT and SD exerted no effect on the Chao1. LUT had a marked influence

on the Shannon ($p = 0.020$) (Figure 2B) and the interaction of LUT and SD ($p < 0.001$) (Figure 2B). In addition, we found that Chao1 and Shannon were significantly influenced by the different land use types at 0–20 cm and 20–40 cm soil depths, respectively (Figure 2). The Chao1 index estimates the richness of bacterial communities, and it was highest (8585.68 ± 114.14) at the soil depth of 0–20 cm in grassland and lowest (4519.41 ± 489.76) at the soil depth of 20–40 cm in *Hylocereus* spp. The Chao 1 index was significantly affected by the soil depth in grassland and *Zea mays*, the Chao 1 index was higher at the soil depth of 0–20 cm than at 20–40 cm, and the Chao 1 index at the soil depth of 0–20 cm was enhanced by 17.3% compared with the soil depth of 20–40 cm in *Zea mays* (Figure 2A). The Shannon index estimates the diversity of bacterial communities, and it was highest (6.63 ± 0.03) at the 20–40 cm soil depth in *Zanthoxylum planispinum* and lowest (6.01 ± 0.09) at the soil depth of 20–40 cm at grassland. The Shannon index was significantly affected by the soil depth in grassland and *Zanthoxylum planispinum*, it was higher at the soil depth of 0–20 cm than 20–40 cm, and at the soil depth of 0–20 cm it was increased by 9.5% compared with the 20–40 cm depth in grassland (Figure 2B). The Shannon index at the soil depth of 0–20 cm decreased by 4.7% compared with 20–40 cm in *Zanthoxylum planispinum* (Figure 2B).

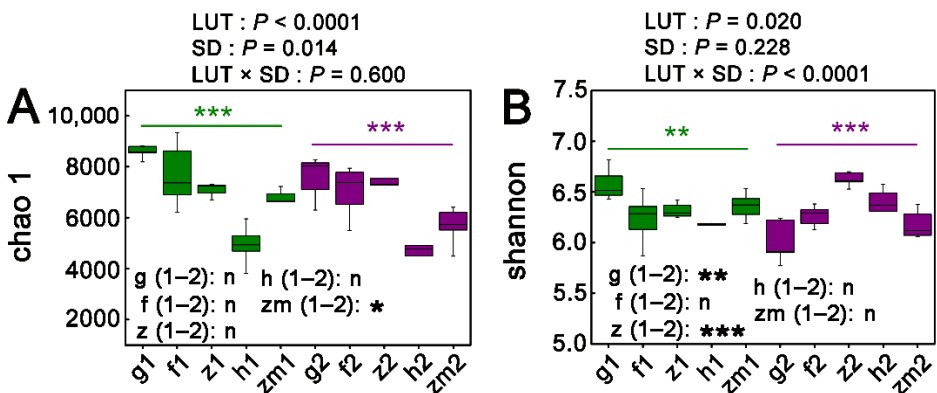

**Figure 2.** Effect of the different types of land use on soil bacterial alpha diversity: (**A**) Chao1, (**B**) Shannon. ** and *** above green line and purple line indicate obvious difference at $p < 0.01$ and $p < 0.001$ at 0–20 cm and 20–40 cm soil depths, respectively, in the five different land use types (ANOVA); n is not statistically significantly different among the five different land use types (ANOVA). *, ** and *** after g/f/z/h/zm (1–2) indicate obvious difference at $p < 0.05$, $p < 0.01$ and $p < 0.001$ between 0–20 cm and 20–40 cm soil depth in the five different land use types, respectively (independent *t*-test), n is not statistically significantly different between 0–20 cm soil depth and 20–40 cm soil depth in the five different land use types, respectively (independent *t*-test). Results of two-way ANOVA for the effects of land use types (LUT) and soil depth (SD) on Chao1 and Shannon. g1: grass land at 0–20 cm; g2: grass land at 20–40 cm; f1: secondary forest land at 0–20 cm; f2: secondary forest land at 20–40 cm; z1: *Zanthoxylum planispinum* land at 0–20 cm; z2: *Zanthoxylum planispinum* land at 20–40 cm; h1: *Hylocereus* spp. land at 0–20 cm; h2: *Hylocereus* spp. land at 20–40 cm; zm1: *Zea mays* land at 0–20 cm; zm2: *Zea mays* land at 0–20 cm.

### 3.3. Absolute Quantification of Soil Bacterial Community

The different land use types (LUT) and soil depth (SD) influenced the major bacterial phyla, but the interaction of LUT × SD only exerted a clear effect on the Actinobacteria (Table 2). At the 0–20 cm depth in the forest, Proteobacteria (28%) and Acidobacteria (26.7%) were the dominant phyla. Acidobacteria (28.4%) and Proteobacteria (21.5%) were the dominant phyla at the 0–20 cm depth of grassland. At the 0–20 cm depth of *Hylocereus* spp., Acidobacteria (25.4%) and Proteobacteria (22.2%) were the dominant phyla. At the 0–20 cm depth in *Zanthoxylum planispinum*, Acidobacteria (29.3%) and Proteobacteria (29.3%) were the dominant phyla. At the 0–20 cm depth in *Zea mays*, Acidobacteria (28.3%) and Proteobacteria (23.9%) were the dominant phyla. In addition, the Actinobacteria, Bacteroidetes, Candidate division, Chloroflexi, Gemmatimonadetes, and Planctomycetes

were also the dominant phyla in the 0–20 cm depth samples under the five different land use types (Figure 3A). Similarly, Acidobacteria and Proteobacteria were the predominant phyla at 20–40 cm in the five different land use types; furthermore, Actinobacteria, Bacteroidetes, Chloroflexi, Gemmatimonadetes, Latescibacteria, Planctomycetes, and Proteobacteria were also the dominant phyla in the 20–40 cm depth samples across the different land use types (Figure 3C).

**Table 2.** Results of two-way ANOVA for the effects of land use type (LUT) and soil depth (SD) on the absolute abundance of Acidobacteria, Actinobacteria, Bacteroidetes, Chloroflexi, Gemmatimonadetes, Planctomycetes and Proteobacteria, F: F-value.

| Treatment | df | Acidobacteria | | Actinobacteria | | Bacteroidetes | | Chloroflexi | | Gemmatimonadetes | | Planctomycetes | | Proteobacteria | |
|---|---|---|---|---|---|---|---|---|---|---|---|---|---|---|---|
| | | F | *p* | F | *p* | F | *p* | F | *p* | F | *p* | F | *p* | F | *p* |
| LUT | 4 | 11.1 | <0.0001 | 27.0 | <0.0001 | 5.4 | 0.001 | 16.3 | <0.0001 | 4.3 | 0.006 | 9.8 | <0.0001 | 8.0 | <0.0001 |
| SD | 1 | 16.5 | 0.0002 | 19.3 | <0.0001 | 22.4 | <0.0001 | 17.7 | 0.0001 | 24.8 | <0.0001 | 7.4 | 0.004 | 9.1 | 0.004 |
| LUT × SD | 4 | 0.8 | 0.545 | 8.1 | <0.0001 | 1.8 | 0.144 | 1.6 | 0.200 | 1.3 | 0.294 | 0.8 | 0.539 | 0.9 | 0.461 |

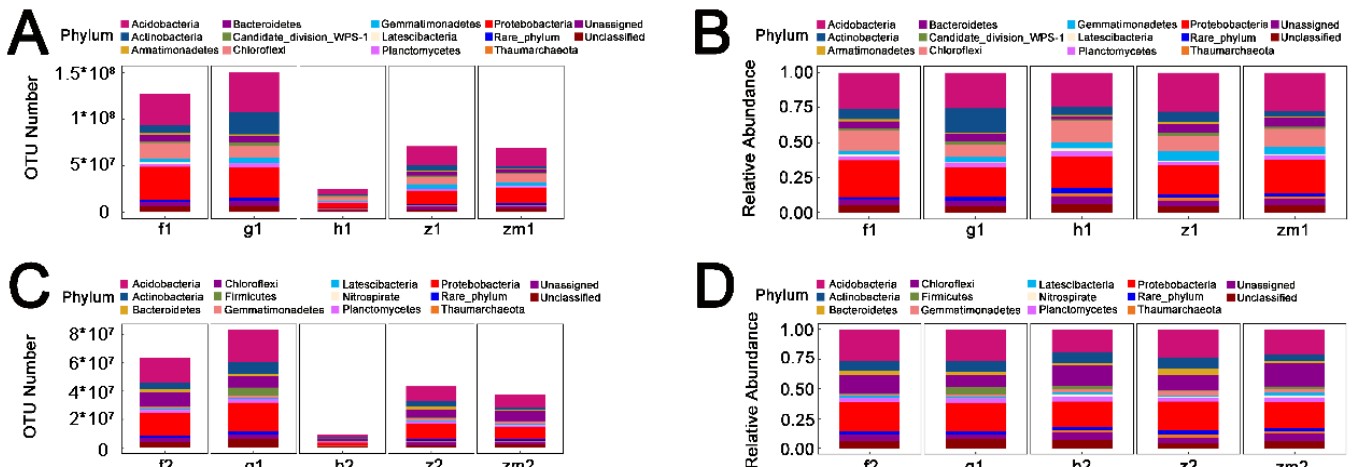

**Figure 3.** Absolute abundances ((**A**,**C**) represent 0–20 cm and 20–40 cm soil depth, respectively, 16S rRNA gene copy numbers per gram of soil) and relative abundances ((**B**,**D**) represent 0–20 cm soil depth and 20–40 cm soil depth, respectively, %) of the major bacterial phyla in all soil samples. g1: grass land at 0–20 cm; g2: grass land at 20–40 cm; f1: secondary forest land at 0–20 cm; f2: secondary forest land at 20–40 cm; z1: *Zanthoxylum planispinum* land at 0–20 cm; z2: *Zanthoxylum planispinum* land at 20–40 cm; h1: *Hylocereus* spp. land at 0–20 cm; h2: *Hylocereus* spp. land at 20–40 cm; zm1: *Zea mays* land at 0–20 cm; zm2: *Zea mays* land at 0–20 cm.

The results of the PCoA (calculated on Bray–Curtis) showed that PCoA1 and PCoA2 explained 35.34% and 17.15% of the total variation in the bacterial communities at 0–20 cm and 20–40 cm across the five land use types. Further, the 0–20 cm and 20–40 cm depths were clearly separate across the land use types. Furthermore, the five land use types were able to significantly separate at 0–20 cm and 20–40 cm depths (Figure 4). In addition, land use type (LUT), soil depth (SD), and the interaction between them exerted a significant ($p = 0.001$) influence on the composition of the bacterial community (Figure 4). At the genus level, the heatmap demonstrated that the clustering pattern of the bacterial communities at depths of 0–20 cm and 20–40 cm was similar to those observed by PCoA (Figure 5). The majority of the top 30 genera at the 0–20 cm depth in grassland and forest demonstrated a higher absolute abundance compared to those of the other land use types (Figure 5). The genera of the bacterial communities of *Hylocereus* spp. at the 0–20 cm soil depth were lowest among the five different land use types (Figure 5). In addition, most of the top 30 genera at the 20–40 cm depth in grassland and *Zanthoxylum planispinum* demonstrated a significantly higher absolute abundance, as shown by the greater number of red columns compared

with the other three different land use types (Figure 5). Most of the top 30 genera at the 20–40 cm depth in *Hylocereus* spp. demonstrated a significantly lower absolute abundance (Figure 5). In addition, our results showed that the five different land use types featured the same 1149 OTUs at 0–20 cm and the same 1103 OTUs at 20–40 cm; however, grass land, *Zanthoxylum planispinum* land, and *Zea mays* land presented no special OTU at the two soil depths (Supplementary Materials, Figure S2).

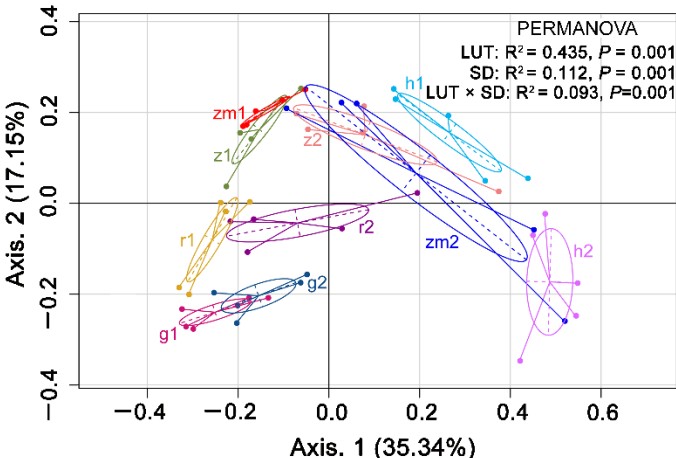

**Figure 4.** Principal coordinate analysis (PCoA) of soil bacterial communities (Bray–Curtis) for all soil samples. g1: grass land at 0–20 cm; g2: grass land at 20–40 cm; f1: secondary forest land at 0–20 cm; f2: secondary forest land at 20–40 cm; z1: *Zanthoxylum planispinum* land at 0–20 cm; z2: *Zanthoxylum planispinum* land at 20–40 cm; h1: *Hylocereus* spp. land at 0–20 cm; h2: *Hylocereus* spp. land at 20–40 cm; zm1: *Zea mays* land at 0–20 cm; zm2: *Zea mays* land at 0–20 cm.

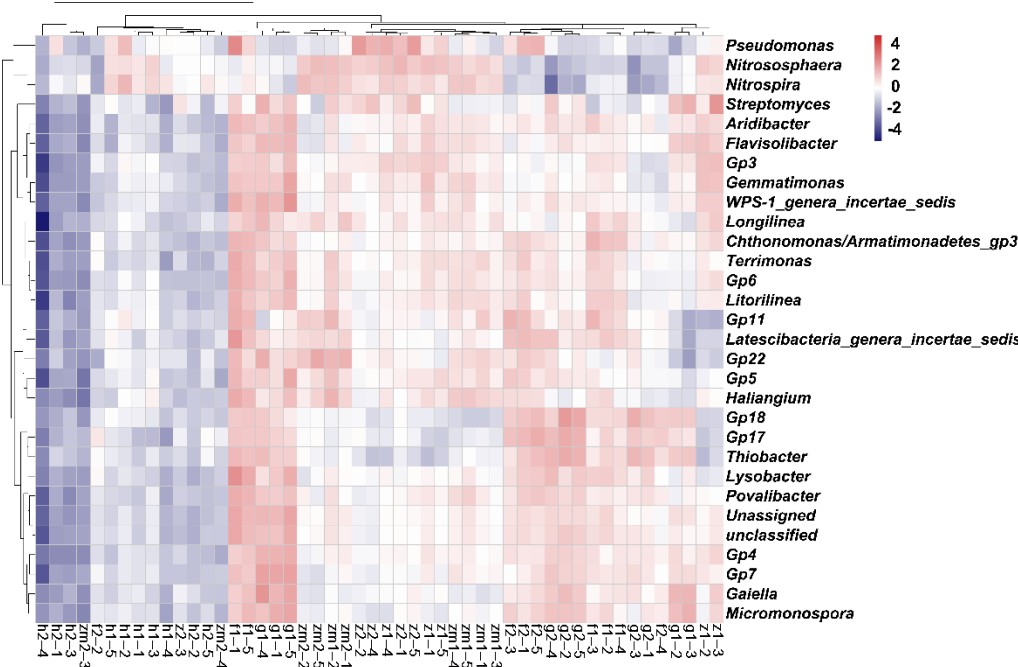

**Figure 5.** The heatmap of the distributions of the 30 richest bacterial genera at 0–20 cm soil depth and 20–40 cm soil depth. The absolute abundances of bacterial genera in soil are indicated by color intensity. g1: grass land at 0–20 cm; g2: grass land at 20–40 cm; f1: secondary forest land at 0–20 cm; f2: secondary forest land at 20–40 cm; z1: *Zanthoxylum planispinum* land at 0–20 cm; z2: *Zanthoxylum planispinum* land at 20–40 cm; h1: *Hylocereus* spp. land at 0–20 cm; h2: *Hylocereus* spp. land at 20–40 cm; zm1: *Zea mays* land at 0–20 cm; zm2: *Zea mays* land at 0–20 cm.

### 3.4. Relevance of Soil Bacterial Communities and Diversity to Soil Nutrients

The results of the Spearman correlation coefficients found a significant relationship between the soil bacterial diversity and soil nutrients at 0–20 cm and 20–40 cm (Figure 6). Further, we found that the Chao1 index changes were closely related to the TN, SOC, and AN ($p < 0.001$); NN ($p < 0.05$) (Figure 6A). Similarly, the Shannon index changes were significantly correlated with the AN ($p < 0.001$), TN, and SOC ($p < 0.05$) (Figure 6A) at the 0–20 cm soil depth. We found that Beta diversity (PCoA) was related to the TN and SOC ($p < 0.001$); AN ($p < 0.05$) at the 0–20 cm soil depth (Figure 6A). Interestingly, at the 20–40 cm soil depth, our results showed that Chao1 was correlated with TN ($p < 0.01$) and pH ($p < 0.001$) (Figure 6B). Shannon was correlated with the SOC ($p < 0.01$), TP, AN, and AP ($p < 0.05$) (Figure 6B). The bacterial Beta diversity (PCoA) was significantly correlated with TN ($p < 0.001$) and pH ($p < 0.001$) (Figure 6B).

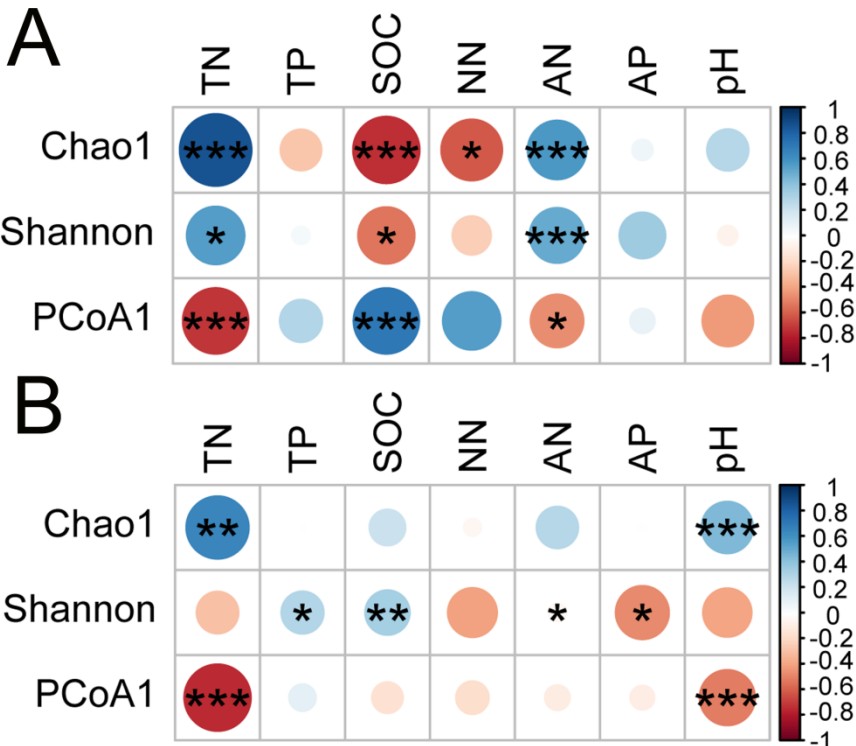

**Figure 6.** Spearman's rank correlation coefficients between soil properties and microbial varieties (Chao1 and Shannon index; PCoA) at 0–20 cm soil depth (**A**) and 20–40 cm soil depth (**B**), respectively. *, ** and *** indicate obvious difference at $p < 0.05$, $p < 0.01$ and $p < 0.001$ between soil properties and microbial Chao1, Shannon index; PCoA.

The results of the RDA (redundancy analysis) showed that the bacterial phyla responded differently to changes in the soil nutrients between the 0–20 cm and 20–40 cm soil depth (Figure 7A,B). The first and second axis of the RDA explain 80.45% and 9.95% of the variance at the 0–20 cm soil depth (Figure 7A). The RDA also highlighted that TN exerted a strong positive effect on Armatimonadetes, Acidobacteria, and Candidate division WPS-1 and a strong negative effect on Thaumarchaeota. AN exerted a strong positive effect on Actinobacteria and a negative effect on Thaumarchaeota. NN exerted a positive effect on Thaumarchaeota and a negative effect on Armatimonadetes and Actinobacteria. SOC exerted a positive effect on all the phyla levels, except Latescibacteria and Thaumarchaeota. The pH exerted a positive effect on Armatimonadetes and Chloroflexi and negative effects on Thaumarchaeota. AP only exerted a negative effect on Latescibacteria. TP exerted no distinct effect on any of the phyla levels at the 0–20 cm soil depth (Figure 7A,C). According to the RDA, TN, AN, and pH were among the soil properties that were important factors

influencing soil bacterial communities in grassland and forest; AP, TP, and NN were the central factors affecting soil bacterial communities in *Zea mays*. SOC was the main factor influencing the soil bacterial community in *Hylocereus* spp. at the 0–20 cm soil depth (Figure 7A).

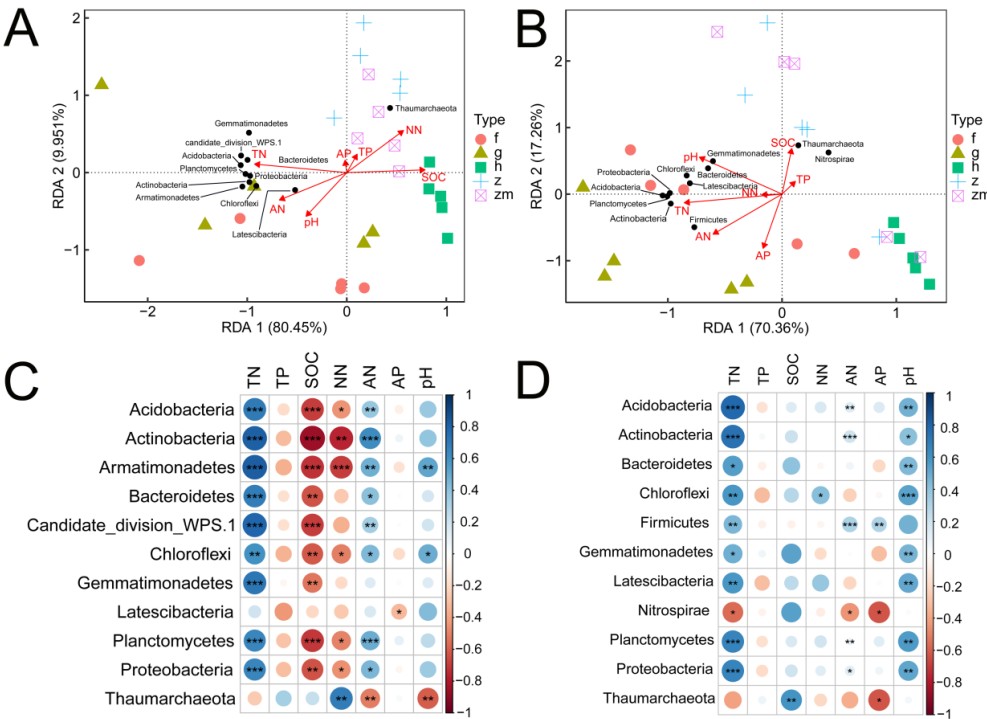

**Figure 7.** Ranking diagram of the redundancy analysis (RDA) to identify the relationship between soil nutrients (red arrows) and the richness of soil microbial phyla (black dots) at 0–20 cm soil depth (**A**) and 20–40 cm soil depth (**B**), respectively; Spearman rank correlation coefficients between soil nutrients and the abundance of soil microbial phyla at 0–20 cm soil depth (**C**) and 20–40 cm soil depth (**D**), respectively. *, ** and *** indicate obvious difference at $p < 0.05$, $p < 0.01$ and $p < 0.001$ between soil nutrients and the abundance of soil microbial phyla.

At the 20–40 cm depth, the first and second axis of the RDA explain 70.36% and 17.26% of the variance (Figure 7B). The RDA also indicated that TN exerted a strong positive effect on Acidobacteria, Actinobacteria, Planctomycetes, and Proteobacteria and a negative effect on Nitrospirae. NN only exerted a positive effect on Chloroflexi. AN exerted a strong positive effect on Actinobacteria and Firmicutes but a negative effect on Nitrospirae. AP exerted a positive influence on Firmicutes and a negative influence on Nitrospirae. Soil pH exerted a strong positive influence on Chloroflexi. TP exerted no marked influence on any of the phyla (Figure 7B,D). Further, TN, AN, AP, NN, and pH were among the soil properties that were the important factors influencing the bacterial community changes in forest and grasslands; SOC and TP were the central factors influencing the bacterial community in *Zea mays* and *Zanthoxylum planispinum* (Figure 7B). Further, according to the variance partitioning analysis (VPA), 17.4% of the variation could be explained by different LUTs, 2.7% of the variation could be explained by SD, 3.6% of the variation could be explained by soil nutrients, 1.9% of the variation could be explained by soil pH, and 1.2% of the variation could be explained by LUT, SD, soil nutrients, and soil pH (Figure 8).

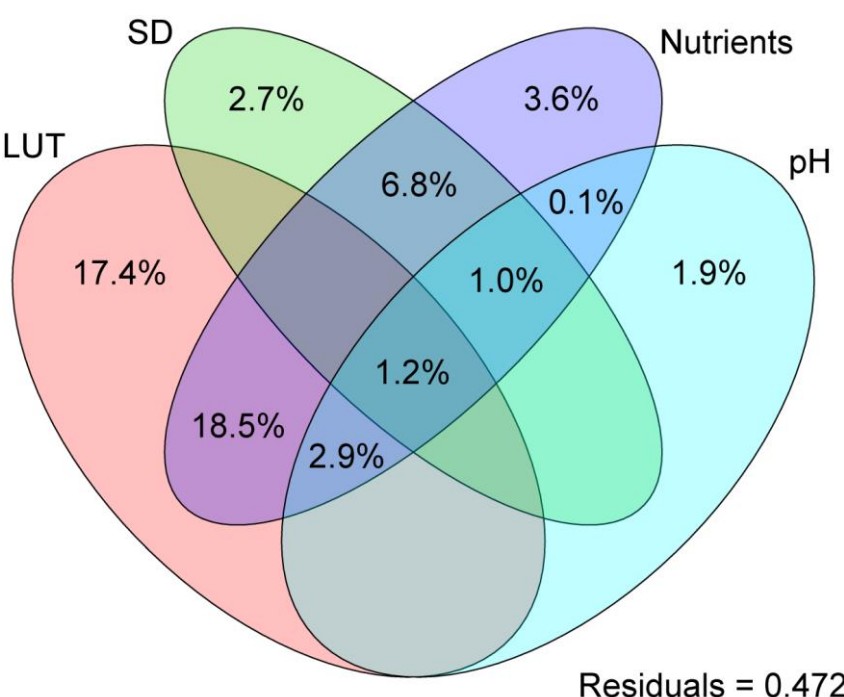

**Figure 8.** Variance partitioning of the proportion of variation in bacterial community that can be explained by nutrient variables (SOC, TN, TP, AN, NN, AP), different land use types (LUTs), and soil depth (SD), as well as residual unexplained variation. Values < 0 are not shown.

*3.5. Structural Equation Model of Soil Nutrients, pH, Land Use Types, and Soil Depth for Bacterial Diversity*

The SEM explained 46% and 65% of the variations in the bacterial community in Shannon and Chao1, respectively, through the LUT and SD (Figure 9). The LUT affected Shannon by influencing SOC ($p = 0.029$) and TN ($p = 0.002$), and SD affected Shannon by influencing SOC ($p = 0.035$, TN ($p = 0.01$), AP ($p = 0.002$), and pH ($p = 0.028$) (Figure 9).

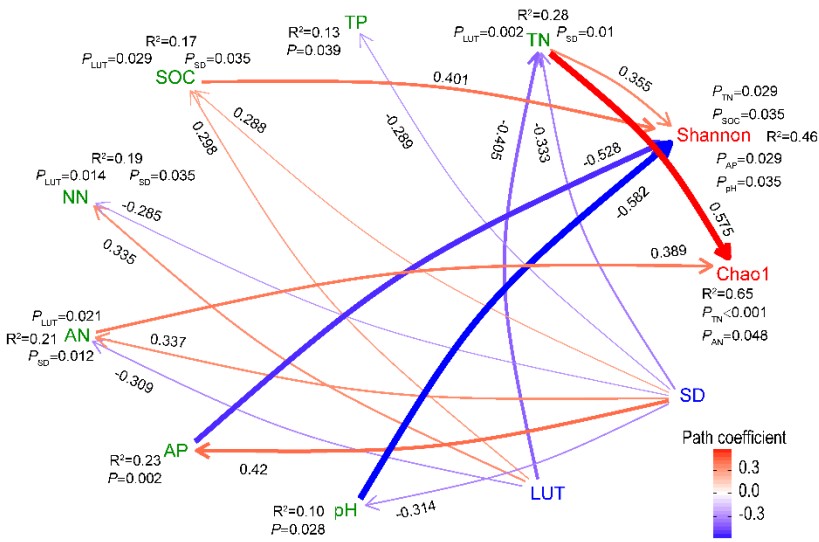

**Figure 9.** Structural equation model (SEM) among soil nutrients, pH, different land use types, soil depth, and bacterial community diversity.

## 4. Discussion

The present study focused on how the land use types and soil depth influenced soil nutrients, pH and soil bacterial community in karst areas, Southwest China. Our results

showed that LUT and SD strongly influenced the diversity and structure of the bacterial community, mainly by affecting the AN and TN content.

Previous studies indicated that land use type and soil depth exert a substantial effect on soil physio-chemical parameters [61–63]. TN and SOC were closely linked and highly correlated with different land use types in karst areas, whilst TN and SOC contents in different land use types increased from agricultural types to forests [7,64,65]. Soil properties are usually regarded as important indicators of soil fertility and can maintain plant productivity [66]. The conversion of forest to pasture strongly influences the structure of soil microbial communities [67]. In addition, the change in land use type significantly influenced TN and SOM contents [68]. Thus, the TN content was highest in grassland among the five different land use types, owing to more litter entering the soil and higher decomposition rates. SOC plays an essential role in soil function, productivity, fertility maintenance, and carbon cycle [7]. It is vital to study the effect of different land use types on the content of SOC in karst areas. Our study found that the SOC content was highest in the 20–40 cm soil depth *Zanthoxylum planispinum* land among the five different land use types. Studies found that SOC content was higher in grassland than other land use types except for natural forest in non-karst areas [69–71]. Changes in SOC content across land use types in the karst area are not consistent with that of non-karst ecotypes. Changes in land use from grassland or forest to arable land or natural vegetation land to agricultural land significantly influence soil nutrient contents [72]. For instance, agricultural practices and tillage influence decomposition and result in the loss of soil nutrients [72]; plants made varying nutritional demands and used nutrients with different efficiencies [73].

Our results also showed that LUT and SD significantly influenced the measured soil parameters. It was reported that the differences in soil TP content might result from an alteration in biogeochemical processes at different soil depths [74,75]. It was further reported that land use types influenced AP, but the interaction of soil depth and land use type was insignificant [76]. Biochemical processes modulate the availability of AP because most of the *p* available is derived from the soil organic matter [77]. The soil pH, TN, and AP were significantly influenced by land use types at non-karst regions [78]. However, our results found that the AP content was highest at the 20–40 cm soil depth in grass land among the five different land use types. It implied that land use play different roles in regulating AP content in karst and non-karst areas. Previous studies focused on depth-specific responses have highlighted that SOC is a significant factor affecting soil subsurface community composition [63]. This suggests that considering that soil pH, temperature, $O_2$ and texture do not generally shift substantially in soil profiles, these parameters likely do not play a major role in defining community structure in soil profiles [32]. Higher AP content in agricultural fields is due to rapid mineralization rates or manure additions [79]. However, our results showed that the TP content was highest at 0–20 cm soil depth in *Hylocereus* spp. land. It was reported that TP content was higher in maize farms compared to that of grassland soils, which could be because of the application of phosphorus fertilizer [80]. It was reported that different land use types significantly affected the alkali-hydrolyzable nitrogen (Nah) content in karst regions, and the Nah content was highest in native forests among shrubland, grassland, and cornfields [81]. Our results demonstrated that the NN content was highest at 0–20 cm soil depth in *Zea mays* land. Some studies reported that the conversion of forest into other land use types could cause soil and water loss [82], decrease the content of TN, TP, and organic soil, modify the soil structure, and reduce the soil quantity [83]. These depth-specific responses were more significant in the different field sites than in the forest sites, despite changes in $NO3^-$, $NH4^+$, and SOC, and showed the same trends in depth [63]. This implies that soil depth strongly influenced $NO3^-$, $NH4^+$, and SOC.

When soil health is recovered, the soil environment is beneficial for microorganisms [84], which results in the faster transformation of soil nutrients [85]. The soil depth and land use type influenced the structure and diversity of soil bacterial communities [32,34,61,86,87], while other studies showed that the different land use practices exert an impact on the structure and diversity

of soil bacterial communities in karst regions [88–90]. Our study showed that LUT and SD significantly influenced alpha diversity patterns; however, SD only significantly affected the Chao1 index. Similarly, a study demonstrated that land use type also influenced the diversity of bacterial communities [91]; notably, it was further reported that the Chao1 and Shannon indices of the bacterial communities were different among woodland, shrubland, and grassland. In addition, different plant species influenced soil microenvironments by root exudates differences or leaf litter difference [92], leading to different structures of bacterial community in different land-use types. Generally, soil pH plays a central role in influencing microbial community distribution [39,93,94]. The Chao1 and Shannon indices at the 0–20 cm soil depth in grassland were higher than those in woodland and shrubland [95]. Another study reported that the diversity of soil bacterial communities was lower in the paddy than those in the corn and citrus lands [88]. A recent study showed that a decrease in the diversity of soil bacterial communities enhances soil depth, which is due to the decrease in the nutrients [96]. These results are consistent with those of other studies about the relationship between soil depth and the diversity of soil bacterial communities [97,98].

The structure of microbial communities was driven by both soil properties and land use types [27,61]. These results are consistent with previous studies, which demonstrated the influence of changes in bacterial community structure on agricultural practices [99]. One study showed that significant changes in the dominant phyla with soil depth influenced the changes in community structure with the soil depth gradient [63]. Our results also showed that Shannon diversity was higher at the 0–20 cm soil depth than at the 20–40 cm soil depth in grassland, forest land, and *Zea mays* land, but lower at the soil depth of 0–20 cm than at 20–40 cm in *Hylocereus* spp. and *Zanthoxylum planispinum*. This may be due to the unique characteristics of karst regions, including poor soils with a high degree of rocky desertification. The PCoA distance demonstrated the significant role of land use types and soil depth in soil bacterial community composition. Our results showed that bacterial communities from the five land use types were distinctly different at the soil depth of 0–20 cm and 20–40 cm, possibly due to the difference in edaphic properties. The bacterial communities of *Zea mays* at 20–40 soil depth were highly dispersed compared to the other land use types, which was probably due to the texture difference of *Zea mays* at 20–40 soil depth. Proteobacteria, Acidobacteria, and Verrucomicrobia were the most abundant dominant bacteria at the soil depth of 0–20 cm in the woodland, shrubland, and grassland [95]; similarly, our results also showed that Acidobacteria and Proteobacteria were the most abundant phyla levels at the soil depth of 0–20 cm and 20–40 cm in the five land use types. Combining the above with our results implies that Acidobacteria and Proteobacteria are important bacteria taxa in the different land use types. Members from the Proteobacteria phyla were observed to decrease with increasing soil depth [31,34,63,100]. It was noted that the phyla levels are dependent on site-specific soil properties [34]; the study showed an increased abundance of the Latescibacteria, Nitrospirae, Gemmatimonadetes, and Chloroflexi phyla with depth [63].

Studies demonstrated that soil nutrients are linked with the diversity and structure of bacterial communities [101,102]. Chao1 and Shannon indices were negatively related with TN and TP content under tobacco–rice rotation; however, SOC, alkalotic nitrogen, and available phosphorus exerted no significant influence on microbial diversity [103]. One study showed that SOC, TN, TP, nitrate nitrogen (NN), and AP were positively correlated with soil bacterial Alpha (Shannon) and Beta diversity. However, ammonium nitrogen (AN) was negatively related to bacterial Alpha (Shannon) and Beta diversity [101]. Another study found that Shannon index changes were closely linked to soil pH under fertilizer management [102]. Bacterial community compositions in different soil depths and different land uses were highly variable.

Interestingly, the present study found that TN and AN were positively correlated with Chao1 and Shannon, but TN and AN were negatively correlated with Beta diversity. By contrast, SOC was negatively related to Chao1 and Shannon, but SOC was positively correlated with Beta diversity at 0–20 cm soil depth in the five land use types. A previous

study found that SOC positively affected bacterial Alpha diversity, which implied that the improvements in carbon source utilization increased microbial diversity under intercropping systems [101]. There were reports that SOC exerted the most significant effect on soil bacterial community structure [104,105]. In general, soil pH played an important role in driving the distribution of soil microbial communities [61,94]. Similar to the results reported by Zhang et al., it was found that Acidobacteria abundance increased with soil pH [106]; however, other studies found that Acidobacteria abundance was negatively related to soil pH [93,94,107], which was inconsistent with our results. The soil of karst systems is often thin and rocky, with relatively high permeability; it is thus difficult to restore when disturbed [6]. Karst systems feature landscapes characterized by sinkholes and caves formed by the dissolution of highly soluble carbonate rock [108]. These unique characteristics of karst regions make it difficult to study how different land use types affect soil nutrients, the diversity of soil microbial communities, and the relationship between them. In addition, it was reported that protecting native forests and inoculation with beneficial microbes is more beneficial for the restoration of karst systems after cultivation [36]. The higher proportion of negatively related OTUs with soil depth for factors influencing microbes across soil nutrient environments, especially, $NO3^-$, $NH4^+$, and SOC, strongly governs the diversity of the microbial community in soils [63,109]. The reported strong effect of pH on shifting taxa distributions in soils [63,110,111] was consistent with our study.

Proteobacteria belong to facultative trophic and aerobic heterotrophic bacteria [112,113], which were found in high SOC in karst regions. One study showed that with an enhancement of soil depth, the contents of soil nutrients reduced and Proteobacteria abundance decreased [95]. Our results showed that Proteobacteria was negatively related with SOC at 0–20 cm soil depth but was positively related to TN and AN at 0–20 cm and 20–40 cm soil depths. Our results showed that TN and AN were more critical for Proteobacteria at 0–20 cm and 20–40 cm soil depths. The previous study showed that Acidobacteria was widespread in various soil types with high abundance [114]. Acidobacteria are acidophilic bacteria, and they could decompose plant and animal residues to form organic carbon [106]. One study showed that Acidobacteria were negatively related to SOC [95]; similarly, our results found that Acidobacteria were negatively related to SOC at the 0–20 cm soil depth. Moreover, due to the complex environment of karst regions, to sum up, we cannot acquire enough information to evaluate the relationship among land use types, soil nutrients, and microbial diversity.

## 5. Conclusions

The present study improved our understanding of how soil depth and land use type influence soil nutrients, the structure and diversity of soil bacterial communities, and the relationship between soil nutrients and soil bacterial community in karst regions. Soil depth and land use type significantly affect soil nutrients and the structure of soil bacterial community, and clear correlations were found. TN, AN, and pH are the key factors driving the variation in soil bacterial communities in grassland and forest; AP, TP, and NN were the essential factors driving soil bacterial community alterations in agricultural land. The effect of land use type was stronger than that of soil depth for the structure of soil bacterial communities. Our results highlighted the different responses of bacterial communities to soil depth and land use type, and shed further light on microbial biodiversity and its ecological role. This provides a theoretical basis for the rational use of limited land in karst areas. Selecting an appropriate land use type according to the structure of soil bacterial communities in karst regions is of important practical significance.

**Supplementary Materials:** The following supporting information can be downloaded at: https://www.mdpi.com/article/10.3390/soilsystems6010020/s1, Figure S1: The rarefaction curve; Figure S2: The OTU venn diagram of five different land use types at 0–20 cm and 20–40 cm, respectively; Table S1: The detail information of five different land use types; Table S2: The information of reads classified as bacteria, archaea, others.

**Author Contributions:** Y.L. (Yuke Li), J.G., J.L. (Jie Liu, liujie791204@126.com), J.W., and Y.Y. designed research; Y.L. (Yuke Li), J.L. (Jie Liu, liujie791204@126.com), J.G., and H.X. performed the research; Y.L. (Yinglong Liu), J.J., C.C., J.W., and W.H. analyzed data; Y.L. (Yuke Li), J.L. ((Jie Liu, jieliu@lzu.edu.cn)), K.M., I.M., and J.W. wrote the paper. All authors have read and agreed to the published version of the manuscript.

**Funding:** This research was financially supported by the Program for the Joint Fund of the National Natural Science Foundation of China and the Karst Science Research Center of Guizhou province (Grant No. U1812401), Changjiang Scholars and Innovative Research Team in University (IRT_17R50), Fundamental Research Funds for the Central Universities (lzujbky-2021-ey01, lzujbky-2021-kb12) in Lanzhou University, the Open Project of State Key Laboratory of Plateau Ecology and Agriculture, Qinghai University (2021-KF-02), Lanzhou University "Double First-Class" guiding special project-team construction fund-scientific research start-up fee standard (561119206), the technical service agreement on research and development of beneficial microbial agents for Alpine Rhododendron (071200001), Guizhou education department program (Qianjiaohe-KY-2018-130), major science and technology sub-project of Guizhou science and technology program (Qiankehe-2019-3001-2).

**Institutional Review Board Statement:** Not applicable.

**Informed Consent Statement:** Not applicable.

**Data Availability Statement:** Not applicable.

**Conflicts of Interest:** The authors declare that they have no competing interests.

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
