# Peer review of "Effects of Different Land Use Types and Soil Depth on Soil Nutrients and Soil Bacterial Communities in a Karst Area, Southwest China"

_soilsystems, doi:10.3390/soilsystems6010020_

Round 1

Reviewer 1 Report

The work by Li and colleagues has as aim the analysis and characterization of the effects of land use type, soil depth and their interactions on edaphic bacterial communities and soil physico-chemical properties (e.g., nutrients). A total of 50 soil samples across five different land use types (i.e., Zanthoxylum planispinum land, Hylocereus spp. land, Zea mays land, grass land and secondary forest land) and two depths (0-20 cm and 20-40 cm) were collected from karst area in Huajiang town, Guizhou province, Southwest China. Total DNA was extracted, and bacterial 16S rRNA gene amplified and sequenced to describe the edaphic bacterial communities associated with different land use and soil depth (n=5 per land use type and depth). As expected, both land use and depth significantly influence the assembly and diversity of bacterial communities. Despite the work presented is interesting, the manuscript should be meliorated and improved.

  • General comment. The manuscript would benefit from proof reading by a native English speaker to meliorate the overall clarity and structure of the different sections. Moreover, table should be revised and formatted (e.g., if p<0.001 present the real p-values, while if it is p<0.0001 it is not necessary), and figures presented in high quality (in man case the written parts are not readable).
  • The results of the study are interesting, but are they that we would expect, or do they provide some novel insights into bacterial community assemblages in soil subject to different use/plant cultivation?
  • The authors should add a small introductive/background part for people are not familiar with the object of this study; i.e., Why it is important study edaphic microbial community? Why compare different land use types? Moreover, the authors did not clearly state their hypothesis and the aim of the work. What do authors expect?
  • Line 29. “….by principal coordinate analysis (PCoA)”. PCoA is not a statistical analysis; the separation of samples in the ordination space should be supported by proper statistical analysis, such as PERMANOVA, adonis, glm. Please revise the text using statistic results to support the findings.
  • Add details regarding the sampling procedures, tools used, plant removal, root removal, sampling close to the plants or between them, etc. Please explain the reason why topsoil was removed (1-2 cm); based on the different plants cultivated/growing in the difference soils and the different ecology of these ecosystems, I suppose that the “top-soils” were different – what these 2 cm content?
  • Can the sampled sediments be considered rhizosphere? Do they present roots? If yes, are the root removed from the samples before DNA extractions?
  • The authors should also add a representative image of the 5 different soils and their location in a map of the region studied. It could be added as supplementary material. As well, it could be useful have a supplementary table that contain information regarding the root system of the different plants growing in the 5 fields; do they reach 40 cm (2nd portion of soil sampled)?
  • How many grams of soils were used to perform the physico-chemical analyses?
  • Total DNA was extracted using power soil kit. Please state it before the sequencing part, as well as specify the method used (e.g., following the methods suggested by the kit) and the quantity of soil used during extraction.
  • Please use “16S rRNA gene” instead of “16S rDNA/rRNA”; check all the text.
  • Which primers were used to amplify the v4-v5 region of 16S rRNA gene? How long is the amplicon? Are 2X250 bp enough to merge forward and reverse reads?
  • The approach of “absolute quantification 16S-seq (HAQS)” is very interesting; please try to better clarify it; if it is necessary add detailed explanation in supplementary methods. I suggest to revise the entire section of methods to add all the details necessary to repeat the experiments/analyses performed here.
  • Please provide rarefaction curve, as well as % of reads classified as non-bacteria (e.g., archaea, chloroplast/mitochondria, unclassified) and not used in the analyses; please add info related to reads specifying the different components.
  • Moreover, the authors never mentioned any use of negative controls. Detection of possible contaminants through the workflow of DNA extraction, PCR amplification and sequencing should be also included in order to remove OTUs from the output results.
  • Specify which version of Silva (or other) database was used to assign taxonomy.
  • Please provide pairwise comparison and not only PERMANOVA main test for betadiversity analysis for both bacteria when possible; please, present these results considering the interaction of the 2 factors when it was significative, or for 1/2 single factors based on p-values obtained with main test (e.g., Table 2, 3).
  • Please use italic font (at least) for bacterial classes, orders, families, genera, and species.
  • Please add a table in which the data of SOC, TN, TP, NN, AN, AP and pH are reported as average and standard deviation (n = 5).
  • Table 1. Among the parameters analysed, only for 2 cases the interaction of the two factors LUT X SD was not significative; please present and discuss the results consistently with the results of statistical analysis; if the interaction is significative, please report and discuss only this. For example, lines 188-192 are not necessary for the parameter affected by the interaction LUT X SD. Revised the entire manuscript based on this comment (refer to any other 2-way PERMANOVA applied).
  • Table 2 is not necessary, add the results in the Figure 2 or in the text (refer to the previous comment).
  • Figure 1. Why legend of the colour/code is panel A? In all the panels of this figure, it will be more useful report the differences across LUT and SD as letters (post-hoc tests) on top of each boxplot, and not as note in the text.
  • Figure 2. It will be more useful report the differences across LUT and SD as letters (post-hoc tests) on top of each boxplot, and not as note in the text.
  • Figure 2. Since the analysis proposed here include the “absolute abundance” of bacterial 16S rRNA gene, I suggest to present these data as additional panel of Figure 2. This will give an idea regarding the bacterial load of the different LUT and SD.
  • The authors could add a Venn diagram analysis to reveal core/specific OTUs across LUT at the different SD, as well as enrichment process (volcano plot or similar) to evaluate change across Sd in each LUT.
  • The authors should substitute Figure 3 with a heat map at phylum level and combine it with Figure 5 to obtained an overall view of taxonomy (4 panels, 2 for phylum and 2 for genus). For each sample, it can be added a bar plot combined with the x-axis to show the absolute abundance in term of reads (like the Figure 3A,C).
  • However, how heatmaps in Figure 5. can show the absolute abundance of genera? The variation of colour range between -3 and 3. Please modify and clarify what you show in these heat-maps. Revise also the caption if it is necessary.
  • Table 4. Why both PERMANOVA and ANOSIM where proposed? Data reported showed different things, because with ANOSIM the interaction was not evaluated. The authors should be consistent with the experimental design proposed and present the correct stat. I also think that this table cnan be removed, and the data reported added within the PCoA graph or in the text of the manuscript since the interaction of LUT and SD is significative. As previously comment, please add also pairwise comparison to evaluate the real differences across group (LUT X SD).
  • Figure 3-5. Along with these figures, it would be quite helpful to have also add the relative abundance for all the bacterial OTUs, preferably in the form of heatmap (supplementary material). This will help the readers to get a complete picture of what is happening across the times in different LUT/SD.
  • Figure 4. Could be interesting discuss and evaluate the dispersion (e.g., permdisp) of group of samples. For examples? Why zm2 is so dispersed compared to the others? Did the authors check for permdisp before to run PRMANOVA? If it has p-values <0.05 this statistical analysis ids not appropriated and other should be applied (manyglm or others).
  • Figure 6B. Is the star for Shannon/AN-correlation correct? It seems to have “0” as correlation.
  • Figure 7. All the analysis were performed considering the interaction LUT x SD, but in this case the data are reported separately for SD. Please, clarify this. Moreover, clearly state which is the dataset plotted in RDA (symbols of types). Are these BC-similarities as in Figure 4? How richness is plotted? Which richness is reported? Is it the number pf OTUs for each phylum? is it based on the absolute abundance? I think this graph/analysis is not clearly explained/reported. Please revise and clarify caption, graphs and text.
  • Combine Figure 6 with Figure 7; since in the figure 7A,B richness (alphadiversity) is reported, I suggest to do a unique analysis considering the relationship between the bacterial community parameters (alpha/beta diversity) and those of soil physio-chemical analyses.
  • Figure 8. “Values < 0 were not shown”. How variation can have values <0%? Are Residuals 4.7%? Please clarify. The sum of all values reported is not 100%; how these % were calculated? From which analysis they originated from? How each parameter was tested?
  • Figure 9. Why interaction among LUT and SD was not included? Panel B and C showed the analysis separately for the two SD, but it should be already considered in the panel A. Why the authors decide to propose also the SEM for each depth? Please clarify this point, and update the text.
  • Discussion should be also revised based on the revision of the results, change in the analyses and correct interpretation of statistical analysis.
  • Please update bibliography

Reviewer 2 Report

The work is very well written and should be published.
Please correct your work:
In the abstract, the abbreviations used should be explained
Figures 1 and 5 are hardly legible
Conclusions should be extended to explain the practical application of the research carried out.

Round 2

Reviewer 1 Report

The authors answered to the comments and they have meliorated the manuscript.